# Modeling Community Residents’ Exercise Adherence and Life Satisfaction: An Application of the Influence of the Reference Group

**DOI:** 10.3390/ijerph192013174

**Published:** 2022-10-13

**Authors:** Huimin Song, Wei Zeng, Tingting Zeng

**Affiliations:** 1College of Tourism, Huaqiao University, Quanzhou 362021, China; 2Business School, Nanfang College, Guangzhou 510970, China

**Keywords:** the reference group theory, life satisfaction, exercise adherence, personal investment, strategic and cultural fit

## Abstract

To expand the application area of the reference group and enrich exercise theoretical research, based on Stimulus-Organism-Response (SOR) framework, this study examines the external factors that motivate adherence to exercise. Taking reference group and strategy and cultural fit as the main stimuli, and personal investment and life satisfaction as mediating variables, this study explores the influence of external stimuli on residents’ exercise behavior. In order to enrich the sample size, two surveys of 734 Chinese residents in two cities (Xiamen vs. Fuzhou) were conducted using factor analyses, regression analysis, and *t*-test analysis. The results indicated that the reference group and strategic and cultural fit as external stimuli impact on residents’ personal investment, life satisfaction and exercise adherence, and that personal investment and life satisfaction as the organism has an impact on residents’ exercise adherence. Personal investment and life satisfaction play a chain mediating role between the reference group and exercise adherence, and between strategy and cultural fit and exercise adherence. Moreover, the *t*-test determined the differences between Xiamen and Fuzhou residents’ exercise adherence and life satisfaction. Residents’ surroundings affect their exercise behavior and life satisfaction. These findings have implications for policymaking aimed at promoting national exercise, which could gradually improve residents’ physical fitness, particularly in light of the current coronavirus emergency.

## 1. Introduction

During the worldwide COVID-19 pandemic, exercise is regarded as an important activity that can lead to residents’ physical health and psychological stability [1], due to its impact on preventing premature death from chronic disease [2]. In addition, physical inactivity increases the worldwide burden of cardiovascular disease as well as premature death by COVID-19 [3]. Thus, regular and consistent exercise is a critical component of living a healthy and long life [4], and it is also suggested that residents with higher life satisfaction are more likely to have healthy habits like exercise adherence [5]. However, as physical inactivity is a worldwide problem, it is necessary to motivate residents to take exercise regularly [6]. Therefore, governments, such as those of China and Australia, pay more attention to the effects of exercise on residents, issue relevant policies, and build exercise infrastructure in communities that has been shown to have certain effects in promoting residents’ adherence to participating in exercise [6].

Exercising is dependent on a variety of elements, including community capability, equipment, transportation, and cost at the policy level, as well as demographic/biological, psychological, and behavioral aspects [7]. For example, because of its effects on people’s evaluation and conduct intention, the reference group would influence their behavior. The reference group is the real or imagined person or group that has a significant influence on individuals’ values, attitudes, norms, and behaviors [8]. With respect to exercise, it could be residents’ family, friends, colleagues, and neighbors. According to Spink, Crozier, and Robinson [9], the perception of the reference group has an impact on teenagers’ efforts to engage in exercise. The reference group serves as a point of comparison in people’s daily lives, as well as a source of knowledge for decision-making, and has particular implications for residents’ behavior [8]. For instance, Pu et al. [10] found reference groups to be a social component that can motivate people to participate in exercises such as golf. Moreover, because motivation is an essential component in residents’ behavior, and the city is both limiting and encouraging to residents’ conduct, it is crucial to delve into the influence mechanism of external variables on residents’ exercise behavior. Stimulus-Organism-Response (SOR) theory explains the formation framework of the individual’s response to external stimuli. As this study explores the effects of external factors on residents’ behavior, SOR theory could explain the framework of the reference group and strategic and cultural fit. Therefore, based on SOR theory, residents’ behavior is originated in individuals’ evaluation of external stimulation, and external stimuli are residents’ behavior influencing factors. However, little comprehensive research exists on how external variables influence people’s willingness to exercise regularly. This study evaluates the contribution of personal investment and life satisfaction as factors influencing the exercise participation of residents, based on SOR theory. Therefore, the general objectives of the study are to: (1) expand the application area of the reference group and enrich exercise theoretical research; (2) investigate the influence of the reference group and strategic and cultural fit on residents’ behavior of exercise adherence; (3) verify the chain mediating effects of personal investment and life satisfaction; (4) ensure the model’s robustness by examining Fuzhou and Xiamen, as well as examine the differences between cities that have distinct city cultures. This study is meaningful for governments to issue useful policies in promoting national health and encouraging residents to exercise.

### 1.1. Literature and Theoretical Background

#### 1.1.1. The Reference Group Theory

Hyman [11] coined the term “reference group” in his findings, indicating that respondents would compare themselves with either individuals or groups. Concerning exercise, this study defines the reference group in the context of the process of individual comparison with other people and groups. The reference group is the real or imagined person or group that has a significant influence on individuals’ values, attitudes, norms, and behaviors, drawing on the research of Mi et al. [8]. The reference group can be thought of as an imagined or real person or group that people want to be a part of (or not) in a given situation, and it can also be seen as the source of norms and values, which also have an impact on people’s views, attitudes, norms, evaluations, perspectives, and behaviors [12,13]. The act of conformity (that is, accepting the behavior of the visible majority in the group) is characterized by the impact of the reference group in behavior science [14]. For instance, social normalcy among friends is a strong predictor of rigorous leisure-time activity [15].

Park and Lessig [16] identified three forms of reference groups: (a) informational; (b) utilitarian; and (c) value-expressive reference groups. The informational influence (II) of the reference group is not about forcing norms or values on consumers but rather about offering consultation and associated information to individuals and then inferring their consumption decision [13]. For instance, travelers obtain trip information and subsequently make associated judgments as a result of communication among reference groups that share travel experiences [17]. Utilitarian influence (UI) is based on conformity. In pursuit of meeting reference groups’ expectations and enhancing self-identity in the reference group, people prefer to act the same as the reference group. For adolescents, the descriptive norms of the group, which include requirements for membership, have an impact on teenagers’ efforts [9]. Value-expressive influence (VEI) is based on the self-identification process. People strive to improve their self-image in the reference group by associating with favorable groups to whom they want to be related or resemble [18]. Friendships are said to be closely linked to adolescent physical activity [19].

Because most exercises are outdoor activities that expose people to the outdoors and other people, the reference group will likely have an impact on people’s exercise behavior. People’s attitudes, emotions, perceptions, and behavior will be influenced and changed by reference groups by communicating knowledge, norms, and values, especially for reference groups that are close and communicate frequently [20]. For instance, social norms have an impact on parents’ and children’s attitudes and behaviors regarding ice hockey [21]. In addition, from the perspective of the external environment of the reference group, immediate social surroundings based on peer pressure intensity, social approval visibility, social contact frequency, and socialization processes all attribute to the effects of the reference group [16]. According to researchers, adolescents’ attitudes and attempts toward exercise are influenced by their friends’ perceptions of exercise [9].

#### 1.1.2. Exercise Adherence

Based on the concept of behavioral intention and loyalty, exercise adherence (EA) is the residents’ behavior of doing exercise regularly and having a high awareness of exercise. Therefore, the intention to adhere to participation is an indicator of exercise continuously [22].

Some studies have indicated that psychology, evaluation, information, perception, cognition, norms, attitude, self-efficiency, and satisfaction are all aspects that appear to influence people’s behavioral intentions [23]. Lu and Su [24] found that the impacts of the reference group, social factors, and extraversion are three major factors that affect customers’ impulse buying behavior. As a source of information, norms, and values, the reference group impacts individuals’ behavioral intentions through their psychology and the desire to be recognized by the reference group [25]. In particular, for the reference group, normative and informational influences are significant in influencing people’s behavioral intention and positively promoting human behavioral intention [13]. Moreover, when products are positioned as value expressive, they elicit consumers’ purchasing intention by affecting consumers’ attitudes towards products [26]. For instance, the reference group will affect adolescents’ attitudes and efforts towards exercise by conveying their perception of exercise [9]. Accordingly, the following hypotheses are proposed:

**Hypothesis** **1** **(H1a).**
*II has positive effects on residents’ EA.*


**Hypothesis** **1** **(H1b).**
*UI has positive effects on residents’ EA.*


**Hypothesis** **1** **(H1c).**
*VEI has positive effects on residents’ EA.*


#### 1.1.3. Life satisfaction

Life satisfaction (SA) refers to residents’ assessment of life, which can help to improve their quality of life and is linked to good mood [27]. Physical exercise is linked to life contentment and happiness for everyone, and healthy behavior is related to life satisfaction [5]. From a bottom-up perspective, life satisfaction depends on some concrete areas in life. We chose the sense of well-being with material life, community life, emotional life, and health/safety as key areas that could measure overall life satisfaction, because the study’s focus is on health care [28].

The reference group not only has effects on residents’ values, evaluation, and norms but also frames residents’ success in terms of life satisfaction [12]. By conveying information, norms, and values, the reference group shapes residents’ cognition to exercise and affects their construction evaluation of life satisfaction. Given that exercise is linked to good aspects of life [5], information, norms, behavior, and value associated with exercise could potentially and positively influence residents’ psychology and then influence their life satisfaction. Accordingly, the following hypotheses are proposed:

**Hypothesis** **2** **(H2a).**
*II has positive effects on residents’ SA.*


**Hypothesis** **2** **(H2b).**
*UI has positive effects on residents’ SA.*


**Hypothesis** **2** **(H2c).**
*VEI has positive effects on residents’ SA.*


Life satisfaction is positively related to residents’ health behavior such as exercise. As residents with higher life satisfaction, they are more likely to adopt advice that is beneficial to life, and more likely to enact healthier behavior which is good for them, in a positive manner [5]. With healthy beliefs, residents tend toward exercise adherence, as exercise is considered an activity that is beneficial to human health.

Accordingly, the following hypothesis is proposed:

**Hypothesis** **3** **(H3).**
*SA has positive effects on residents’ EA.*


#### 1.1.4. Personal Investment

Personal investment means that individuals would invest their limited resources in the goal they want to achieve [29]. Personal investment theory is a motivation theory and contains three components: facilitating condition, sense of self, and perceived goals [30]. People will decide based on their resources the goal that they want to achieve, their social goals, and their social performance/context, which are all motivations for people to take action and they will be affected by their relatives’ support [30]. In other words, personal investment (PI) means that people would invest their limited resources in one thing to achieve a certain purpose in a certain situation, and this reflects personal resources that are put into the activity, such as time, effort, and energy, which would be lost and cannot be recovered if participation is discontinued [29].

As social performance/context could determine residents’ personal investment, the reference group which is part of individuals’ sociological environment has an impact on residents’ cognitive and personal investment in exercise. Therefore, the reference group could affect residents’ personal investment, to its effects on shaping residents’ cognition and value. The following hypothesis is proposed:

**Hypothesis** **4** **(H4a).**
*II has impacts on residents’ PI.*


**Hypothesis** **4** **(H4b).**
*UI has impacts on residents’ PI.*


**Hypothesis** **4** **(H4c).**
*VEI has impacts on residents’ PI.*


Findings have indicated that personal investment affects residents’ exercise behavior [31] based on their limited resources and the goal they want to achieve in a certain situation. Furthermore, personal investment, which relates to residents’ psychological component, could affect their life satisfaction in terms of their sense of fulfillment [31]. The following hypotheses are proposed:

**Hypothesis** **5** **(H5a).**
*PI has positive effects on residents’ EA.*


**Hypothesis** **5** **(H5b).**
*PI has positive effects on residents’ SA.*


#### 1.1.5. Strategic and Cultural Fit

Strategic and cultural fit (FIT) refers to the concept that the activity held by the city should suit the city’s development, and involves coordination and promotion of the city’s image and strategy [32]. As China is promoting national fitness in every city, in this study strategic and cultural fit is more likely the social support for national fitness, such as promotion and infrastructure. Consistent with Jago, Chalip, Brown, Mules, and Ali [33], strategic and cultural fit is defined in this study as a main latent variable that could be measured by four manifest variables: capacity, values, infrastructure, and media. Capacity refers to the host city’s capability to plan and organize an event that reflects the core values of the event. Values relate to the benefits of hosting the event. Infrastructure mainly includes the transportation and exercise infrastructure for residents. Media was included in this construct since it is a major indicator of the fit between a special mega-event and the host city [34].

Strategic and cultural fit is an action that a city adopts to promote an event or an activity, which could shape residents’ attitudes, perceptions, and behaviors [35], and improve residents’ life satisfaction [36]. For example, research on health promotion behavior in South Korea found that social support (referring to capability and infrastructure) affects health promotion behavior and moderates the relationship between self-efficiency and health promotion [37].

Accordingly, the following hypotheses are proposed:

**Hypothesis** **6** **(H6a).**
*FIT has positive effects on residents’ PI.*


**Hypothesis** **6** **(H6b).**
*Italic>FIT has positive effects on residents’ EA.*


**Hypothesis** **6** **(H6c).**
*FIT has positive effects on residents’ SA.*


#### 1.1.6. Mediating Effect of Personal Investment and Life Satisfaction

Stimulus-Organism-Response (SOR) framework suggests that stimuli elements affect individuals’ psychology, evaluation, and cognition towards things, which then influence individuals’ responses [38]. For instance, Jang, Byon, and Yim [39] found that sportscape has an impact on individuals’ emotions as spectators, which then affects their behavior intention. Emotion is found to exert mediating effects between sportscape and behavior intention, based on a SOR framework. The reference group and strategic and cultural fit contain the most external stimuli for residents, which all have an impact on residents, and which would change residents’ attitude towards exercise as well as the investment in exercise. Furthermore, as residents invest more in exercise, under the influence of the reference group and strategic and cultural fit, residents’ psychological fulfillment and life satisfaction will be higher, at which point residents are more likely to exhibit adherence to exercise. Therefore, in SOR theory, the reference group and strategic and cultural fit are the stimuli that have an impact on residents’ psychology, cognition and evaluation, personal investment; life satisfaction could be seen as an organism in SOR theory; exercise adherence is the residents’ response to the stimuli. In accordance with this, under the influence of the reference group and strategic and cultural fit, residents’ personal investment and evaluation of life have changed, and the response will also be different. Therefore, personal investment and life satisfaction play a chain mediating role between the reference group and exercise adherence, strategic and cultural fit, and exercise adherence. Accordingly, the following hypotheses are proposed:

**Hypothesis** **7** **(H7a).**
*PI and SA have a positive chain mediating effect between II and EA.*


**Hypothesis** **7** **(H7b).**
*PI and SA have a positive chain mediating effect between UI and EA.*


**Hypothesis** **7** **(H7c).**
*PI and SA have a positive chain mediating effect between VEI and EA.*


**Hypothesis** **7** **(H7d).**
*PI and SA have a positive chain mediating effect between FIT and EA.*


To present the relationships between invariable, this research study has developed a coceptual framework in Figure 1.

## 2. Materials and Methods

### 2.1. Introduction of Xiamen and Fuzhou

Xiamen and Fuzhou are both in China’s Fujian Province. Fuzhou and Xiamen have the basics of physical infrastructure, and national exercise is a core value in city development. In Xiamen, the government keeps schools open for the public, so residents could have more places to exercise. On 22 July 2021, the Xiamen center information website showed that after 88 schools opened to the public, there were more than 2.33 million residents who exercised in schools in 2021. In Fuzhou, as shown on the Fuzhou sports bureau website on 19 February 2021, the government tried to create ten-minute access to exercise to provide convenience for residents for exercise. The government’s sports website showed that 425 exercise paths have been refreshed or replaced and 80 reconstruction projects have been completed in 2020. Moreover, more than 52% of the residents in Fuzhou usually engage in exercise, as per the data shown in government sports bureaucracy statistical data for 2020. Therefore, Fuzhou and Xiamen are appropriate cities to conduct exercise research, as these two cities have carried out many efforts in promoting national exercise. There are also differences between Xiamen and Fuzhou. Xiamen is a tourist city with a city culture of leisure, and Fuzhou is a city with a weaker leisure culture. Moreover, the urban development is not the same between Xiamen and Fuzhou, as Xiamen has developed further than Fuzhou, with a higher GDP per capita.

### 2.2. Construction Measurement

The influence of the reference group is measured by informational influence, utilitarian influence, and value-expressive influence, and the scale has a total of 13 items [40]. Personal investment is measured by the willingness of people to invest time, energy, and money in exercise, with a total of 3 items [29]. Strategic and cultural fit is an indicator of a city’s support for exercise and is measured by capability, infrastructure, value, and media, with a total of 12 items [33]. Life satisfaction includes a sense of well-being with material life, community life, emotional life, and health/safety, with a total of 9 items [28]. Exercise adherence is measured by the intention toward exercise adherence, with a total of 4 items [41]. Demographic variables included sex, age, economic income, educational level, and past exercise experience. The questionnaire contains 44 items, and the measurement uses the 5 point Likert scale, ranging from 1 (strongly disagree) to 5 (strongly agree).

### 2.3. Reliability

A pilot test was conducted to test the internal consistency of the questionnaire items. The first draft of the survey instrument was distributed to 80 randomly-selected exercise participants who exercised in Xiamen (40) and Fuzhou community exercise centers (40). A total of 72 completed surveys were returned. A reliability analysis (Cronbach’s alpha) was performed for Informational Influence, Utilitarian Influence, Value-expressive Influence, Personal Investment, Strategic and Cultural Fit, Life Satisfaction, and Exercise Adherence, resulting in robust α values of 0.896, 0.932, 0.901, 0.886, 0.911, 0.935 and 0.929, respectively. An alpha of 0.7 or above is considered acceptable as a good indication of reliability [42]. Based on the results of the pilot test and feedback from participants, the final version of the survey instrument was developed (see Table 1).

### 2.4. Data Collection

Investigators sent questionnaires to Fuzhou and Xiamen and delivered questionnaires both online and offline in consideration of COVID-19. We sent an offline questionnaire to exercise places for voluntary completion and collected online questionnaires from the people who live in Fuzhou and Xiamen by people who have been surveyed. For online questionnaires, in attempting to guarantee the quality of the sampling, a sample that took more than 60 s to complete would be accepted, and any incomplete survey, as well as those exhibiting logical problems, would not be included. After filtering, 734 samples were collected in total, 411 samples in Fuzhou and 323 samples in Xiamen with response rates of 90.32% and 88.56%, respectively.

### 2.5. Data Description

Female residents outnumbered their male counterparts (52.10% female vs. 47.80% male in Xiamen, 56.20% female vs. 43.80% male in Fuzhou), and the majority of residents were between the ages of 31–40 in Xiamen (46.20%) and 19–30 in Fuzhou (35.50%). In addition, most residents are well educated in Xiamen (with 73.10% having at least a bachelor’s degree) and affluent (with 65.80% having an income over 6000 ¥), which is contrary to Fuzhou, which has 68.10% holding qualifications lower than a bachelor’s degree, and less income (with 48.70% having less than 4000 ¥ income).

## 3. Results

### 3.1. Confirmatory Factor Analysis

AMOS 24 was used to assess the structural validity of the reference group. The model fit indices in this study are acceptable both in Fuzhou (χ2/df = 1.876, IFI = 0.983, CFI = 0.938, TLI = 0.931, RMSEA = 0.046) and Xiamen (χ2/df = 2.104, IFI = 0.930, CFI = 0.929, TLI = 0.922, RMSEA = 0.056), with all model fit indices having an acceptable range of good fit [42].

As shown in Table 1, all factor loadings exceeded 0.6, except for UI2 in both Fuzhou and Xiamen. As the content of UI2 is essential in assessing the utilitarian influence in the context of national exercise, this study chose to reserve this item. The threshold of composite reliability of all constructs is 0.7, and all values exceeding 0.7 indicates that the questionnaire is reliable [43]. Cronbach’s alpha reflects the scale’s internal consistency, as all Cronbach’s alpha values exceed 0.7, which indicates high internal consistency. The threshold of average variance extracted (AVE) is 0.5, and all constructed exceed 0.5, which indicates high convergent validity [44]. The entire estimated factor loading exceeds the threshold of 0.5 (*p* < 0.001), which indicates the high validity of the scale.

To evaluate the discriminant validity of the scale, the square roots of the average variance are extracted and the interconstructed correlation are compared. As shown in Table 2, all the squares of the average variance extracted value exceed the interconstructed correlation value, which indicates good discriminant validity.

### 3.2. t-Test Analysis

Table 3 shows that there are differences in II, VEI, and PI between Xiamen and Fuzhou. For residents in Xiamen, the value of II is lower than in Fuzhou, and the value of UI and PI is higher than in Fuzhou. Also, residents in Xiamen have higher levels of both education and income than those in Fuzhou, which is consistent with urban development. Regarding exercise experience and frequency, Xiamen has a longer exercise experience with a lower exercise frequency, which may be consistent with the fact that Xiamen has a stronger leisure culture and residents have a strong exercise consciousness.

### 3.3. Hypothetical Path Analysis

Model 4 of process and linear regression in SPSS was applied to examine the impacts of the reference group, personal investment, and strategic and cultural fit on exercise adherence, as well as life satisfaction. Table 4 shows the regression results after controlling for the variables (experience, frequency, gender, sex, education, income) of the process. As results show, reference group and strategic and cultural fit all have impacts on exercise adherence, life satisfaction, and personal investment in both Fuzhou and Xiamen. Additionally, personal investment has an impact on life satisfaction and exercise adherence. Finally, life satisfaction is the predictor of exercise adherence. Thus, H1a, H1b, H1c, H2a, H2b, H2c, H3, H4a, H4b, H4c, H5a, H5b, H6a, H6b, and H6c are not rejected.

### 3.4. Mediation Analysis

Bootstrapping was used to test the mediating effects of life satisfaction. To obtain the 95% confidence intervals (CIs) [45], 5000 bootstrap samples were applied to estimate the direct and indirect effects [46]. Table 5 presents the effects and 95% CIs for II-PI-SA-EA, UI-PI-SA-EA, VEI-PI-SA-EA, and FIT-PI-SA-EA, both in Fuzhou and Xiamen. As 95% CIs do not contain 0, the reference group affects exercise adherence through life satisfaction, as well as personal investment and strategic and cultural fit. Therefore, H7a, H7b, H7c, and H7d are not rejected.

## 4. Discussion

The results show that the proposed model fits both Xiamen and Fuzhou, which stresses that the model is reliable. There are some differences between Xiamen and Fuzhou. Consistent with formal research, city culture and urban development have impacts on residents’ values and behaviors and are related to residents’ exercise awareness. For residents in Xiamen, the value of II is lower than that in Fuzhou, and the value of PI and UI is higher than that in Fuzhou. This indicates that, as Xiamen is a leisure city with a higher level of urban development, residents have a strong awareness of exercise and are more likely to invest in themselves and be driven by social factors that belong to a certain group. With a strong exercise adherence, residents would search for related information by themselves rather than in the reference group. Moreover, with higher education levels, more income, and urban development, residents are more likely to exercise regularly, which is consistent with formal research [7]. Therefore, there are significant differences in II, PI, and UI between Xiamen and Fuzhou. Furthermore, with more residents taking exercise as adherence, the utilitarian influence of the reference group is higher with conformity and the possibility of exclusion from the group.

The results also showed that the reference group has an impact on residents’ exercise adherence, life satisfaction, and personal investment in both Xiamen and Fuzhou. In line with the research of Ding et al. [13] that informational and utilitarian reference groups have impacts on individuals’ purchasing behavior, it is supported that informational reference groups affect residents’ personal investment, life satisfaction, and exercise adherence. When residents exercise, they are more likely to gather information related to exercise from the reference group, which contains the message that exercise is good for health, and regular exercise is seen as a healthy lifestyle, so that residents invest more in exercise. In addition, this exercise-related information also guides people to exercise in the right way without physical injury. Exercise is taken as a norm in daily life, as it is affected by the informational reference group, and it could improve residents’ health and their life satisfaction. For utilitarian influence, based on conformity (avoiding punishment or gaining reward), residents’ behavior and attitude towards exercise would be affected. It could be inferred that individuals would recognize the effectiveness of exercise and cater to the reference group by raising personal investment and exercise adherence for gaining reward (such as admiration of exercise) or avoiding punishment (such as being sneered at for bad physical condition by the group or excluded from the group). Furthermore, expected rewards and improved health lead to residents’ life satisfaction. In contrast to the research of Mi et al. [8] that the value-expressive reference group inhibits individuals’ low-carbon behavior, the results show that the value-expressive reference group influences values. Under the influence of the value-expressive reference group, residents try to enhance their self-image and self-identity among the reference group and become involved in the reference group in terms of belonging to a certain group by exercising adherence and regularity. Moreover, by conveying the positive value of exercise, the value-expressive reference group positively affects residents’ personal investment and life satisfaction. Overall, the reference group could be a factor that motivates residents to take exercise, and then affects their personal investment and life satisfaction.

In addition, strategic and cultural fit is a form of social behavior that supports holding events in the city and will influence residents’ exercise, which is inconsistent with the formal findings [47] that social support could affect residents’ behavior. Currently, as most civilians live in the city, by creating an atmosphere in the city and providing necessary infrastructures for residents, strategic and cultural fit makes residents feel that exercise is necessary, and thus urges them to invest more in exercise and adherence to exercise. As strategic and cultural fit meets residents’ exercise need and shows the capability of the city to meet residents’ needs for a better life, it could infer that residents’ attitudes towards life would be better.

As Tappe and Duda [31] supported, personal investment has an impact on residents’ exercise behavior, which is a motivation in people’s daily lives through allocating limited resources to reach a goal. Exercise requires time, energy, and money, which are opportunity costs in resident life. Under the influence of the reference group and strategic and cultural fit, residents would invest more in exercise. Therefore, as residents put these resources into exercise, they would be more likely to exercise adherence to reach the goal of being fit or other kinds of goals, as well as not to waste the resources that have been invested in the early stage. As they invest more resources in exercise, and the states of their psychological component improve, they will be more satisfied with their life [31] with healthier bodies.

Furthermore, the results draw the same conclusion as Wiese et al. [5], which indicated that life satisfaction is related to healthy behavior: with higher life satisfaction, residents are more likely to exhibit adherence to exercise. As residents with higher life satisfaction, they are more likely to take on positive activities, ensuring life is good. Exercise is considered a healthy behavior that is good for people’s physical and psychological well-being, so residents are more likely to display exercise adherence with higher life satisfaction.

Finally, the results show that the reference group and strategic and cultural fit as “stimulations” will affect exercise adherence in exercise through personal investment and life satisfaction. Based on SOR theory, the reference group and strategic and cultural fit as stimulations will affect residents’ investment in exercise, then their evaluation of life, and finally affect their exercise adherence behavior. The reference group and strategy and cultural fit as external stimuli could affect residents’ attitude towards exercise, make residents know the importance and benefits of the exercise, to change personal investment in exercise. Then, as residents invest more in exercise, psychological fulfillment leads to residents’ life satisfaction. Finally, residents are more likely to participate in the exercise. Therefore, personal investment and life satisfaction exert important roles in the process on the path of these antecedents to exercise adherence.

## 5. Conclusions

### 5.1. Theoretical Implications

First, this study sheds light on facilitating residents’ exercise adherence. Most studies focus on barriers and facilitation of participation in exercise, and take personal characteristics as an important factor. This study takes the reference group and strategic and cultural fit as the external factors that exert the most influence on residents’ daily lives. Extending the exercise research, this study delves into the source of value and norms for people to continue to exercise [7] from the perspective of social relationships, revealing the framework of the reference group’s impacts on residents’ exercise behavior, which is the same as the research that applied the concept of the reference group to consumer behavior.

Second, unlike the previous study that paid attention to the importance of strategic and cultural fit in shaping city brands [32], this study aimed to survey the impacts of strategic and cultural fit on residents’ behavior in promoting citywide exercise activity, revealing the important role of the city in residents’ behavior, especially for exercise behavior.

Third, this study combines external and internal influencing factors, explores the residents’ behavior formation framework based on SOR theory, and verifies the chain mediating effects of personal investment and life satisfaction.

### 5.2. Managerial Implications

This study has some implications for promoting national exercise. First, residents need to exercise in their daily lives, as it not only improves their physical and psychological health but also improves their overall life satisfaction, which is better for social construction and development. Therefore, governments need to encourage residents to exercise, and implement policies to promote national exercise. The government may need to provide professional guidance to residents to lead them to exercise in the right way, as it is essential in residents’ decision making. Moreover, the government could organize activities for residents to exercise, as residents would exercise to show that they belong to the group. Thus, it would be better for governments to allow the existence of various interest groups and to promote exercise through these groups or establish more exercise organizations to promote the development of exercise and to promote national exercise.

Furthermore, affordability is an important factor, as the personal investment which contains time, money, and effort invested in exercise could affect residents’ life satisfaction and exercise adherence, so reducing the cost of exercising is necessary to encourage people to exercise. Reduce the per exercise cost, and in the long term residents would find they have invested more than they thought and will not give up easily.

Last, improve the city’s ability to provide exercise to residents and encourage people to exercise: set more infrastructure, convey the concept of exercise to the public through promoting more exercise information in city media, provide more convenience for residents to exercise, hold more activities for people to exercise taking the community as a unit, and include exercise-related information in the city’s promotional videos, making exercise a part of the city’s image, which all enhance promoting national exercise.

## 6. Limitations and Future Research

There are limitations to this research. For instance, this study does not take the influence degree of the reference group into consideration. For instance, families and friends may have different degrees of influence. Future research could explore the extent to which different reference groups, such as neighbors, family, colleagues, and friends, influence people’s exercise behavior.

## Figures and Tables

**Figure 1 ijerph-19-13174-f001:**
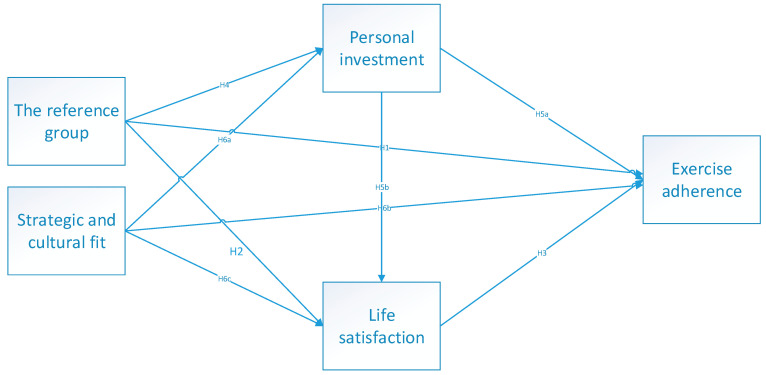
Conceptual model.

**Table 1 ijerph-19-13174-t001:** The results of confirmatory factor analysis.

	Fuzhou				Xiamen			
Items	St FL	CR	AVE	Cronbach’s α	St FL	CR	AVE	Cronbach’s α
II		0.813	0.522	0.812		0.816	0.531	0.805
II1	0.719				0.528			
II2	0.777				0.748			
II3	0.701				0.815			
II4	0.690				0.789			
UI		0.744	0.496	0.726		0.786	0.562	0.749
UI2	0.574				0.518			
UI3	0.802				0.847			
UI4	0.718				0.836			
VEI		0.828	0.547	0.827		0.833	0.558	0.826
VEI1	0.720				0.767			
VEI2	0.753				0.822			
VEI3	0.787				0.621			
VEI4	0.695				0.762			
PI		0.855	0.664	0.842		0.895	0.741	0.886
PI1	0.844				0.892			
PI2	0.880				0.944			
PI3	0.711				0.732			
FIT		0.917	0.503	0.920		0.940	0.590	0.943
FIT1	0.715				0.771			
FIT2	0.696				0.794			
FIT3	0.737				0.816			
FIT4	0.771				0.740			
FIT5	0.714				0.700			
FIT6	0.699				0.743			
FIT7	0.708				0.833			
FIT8	0.708				0.814			
FIT9	0.701				0.814			
FIT10	0.708				0.744			
FIT11	0.634				0.658			
SA		0.900	0.530	0.902		0.913	0.570	0.915
SA1	0.673				0.610			
SA2	0.670				0.702			
SA3	0.696				0.694			
SA4	0.749				0.732			
SA5	0.763				0.715			
SA6	0.740				0.870			
SA7	0.764				0.856			
SA8	0.762				0.824			
EA		0.864	0.615	0.861		0.892	0.676	0.887
EA1	0.792				0.864			
EA2	0.819				0.821			
EA3	0.804				0.894			
EA4	0.717				0.695			

Notes: II = informational influence, UI = utilitarian influence, VEI = value-expressive influence, PI = personal investment, FIT = strategic and cultural fit, SA = life satisfaction, EA = exercise adherence.

**Table 2 ijerph-19-13174-t002:** Discriminant validity matrix.

Constructs	City	II	UI	VEI	PI	FIT	SA	EA
II	Fuzhou	0.722						
	Xiamen	0.729						
UI	Fuzhou	0.449	0.704					
	Xiamen	0.430	0.750					
VEI	Fuzhou	0.525	0.501	0.740				
	Xiamen	0.454	0.522	0.747				
PI	Fuzhou	0.481	0.526	0.532	0.815			
	Xiamen	0.413	0.555	0.556	0.861			
FIT	Fuzhou	0.570	0.428	0.522	0.510	0.709		
	Xiamen	0.587	0.536	0.546	0.545	0.768		
SA	Fuzhou	0.550	0.523	0.494	0.507	0.596	0.784	
	Xiamen	0.530	0.426	0.536	0.568	0.519	0.822	
EA	Fuzhou	0.481	0.555	0.487	0.546	0.503	0.570	0.728
	Xiamen	0.478	0.499	0.595	0.585	0.575	0.521	0.755

**Table 3 ijerph-19-13174-t003:** Results of *t*-test.

Variance	City	Mean	F	sig	t	Sig.	Results
II	Xiamen	3.839	4.818	**	−3.101	***	Xiamen < Fuzhou
	Fuzhou	4.007					
UI	Xiamen	4.289	4.431	*	2.861	***	Xiamen > Fuzhou
	Fuzhou	4.156					
PI	Xiamen	3.923	0.860	n.s	2.838	***	Xiamen > Fuzhou
	Fuzhou	3.742					
Experience	Xiamen	2.260	43.014	***	2.991	***	Xiamen > Fuzhou
	Fuzhou	2.020					
Frequency	Xiamen	1.840	1.249	n.s	−5.269	***	Xiamen < Fuzhou
	Fuzhou	2.160					
Education	Xiamen	2.850	25.422	***	15.237	***	Xiamen > Fuzhou
	Fuzhou	1.880					
Income	Xiamen	3.370	82.122	***	15.039	***	Xiamen > Fuzhou
	Fuzhou	1.920					

Note: * *p* < 0.05; ** *p* < 0.005; *** *p* < 0.001, n.s > 0.05.

**Table 4 ijerph-19-13174-t004:** Empirical results.

Variable		Xiamen				Fuzhou			
		β	S.E.	t	Results	β	S.E.	t	Results
EA	II	0.366 ***	0.047	8.151	support	0.544 ***	0.045	12.183	support
	UI	0.347 ***	0.056	7.619	support	0.380 ***	0.052	9.036	support
	VEI	0.471 ***	0.045	11.016	support	0.318 ***	0.044	7.313	support
	PI	0.560 ***	0.042	13.053	support	0.473 ***	0.036	11.760	support
	FIT	0.480 ***	0.049	11.478	support	0.549 ***	0.051	14.021	support
	SA	0.647 ***	0.044	17.084	support	0.680 ***	0.043	18.772	support
SA	II	0.447 ***	0.044	9.628	support	0.450 ***	0.040	11.392	support
	UI	0.307 ***	0.054	6.144	support	0.381 ***	0.046	8.618	support
	VEI	0.445 ***	0.045	9.410	support	0.362 ***	0.038	8.069	support
	PI	0.508 ***	0.042	10.411	support	0.506 ***	0.031	12.165	support
	FIT	0.625 ***	0.042	15.548	support	0.615 ***	0.043	15.643	support
PI	II	0.385 ***	0.054	7.817	support	0.499 ***	0.054	11.660	support
	UI	0.471 ***	0.059	10.007	support	0.429 ***	0.063	9.595	support
	VEI	0.499 ***	0.051	10.795	Support	0.470 ***	0.050	10.755	support
	FIT	0.523 ***	0.056	11.498	support	0.491 ***	0.063	11.404	support

Note: ***: *p* < *0*.001.

**Table 5 ijerph-19-13174-t005:** Mediating effect test.

Effect Type	Xiamen				Fuzhou			
	Coefficient	S.E.	Bootstrap 95%CI		Coefficient	S.E.	Bootstrap 95%CI	
			LLCI	ULCI			LLCI	ULCI
II→PI→EA	0.151	0.025	0.103	0.202	0.103	0.028	0.049	0.159
II→SA→EA	0.145	0.028	0.093	0.204	0.164	0.030	0.110	0.227
II→PI→SA→EA	0.069	0.015	0.043	0.100	0.080	0.016	0.051	0.114
UI→PI→EA	0.169	0.029	0.113	0.228	0.097	0.025	0.051	0.146
UI→SA→EA	0.065	0.022	0.023	0.110	0.121	0.027	0.071	0.177
UI→PI→SA→EA	0.098	0.018	0.066	0.138	0.091	0.018	0.059	0.130
VEI→PI→EA	0.173	0.028	0.120	0.229	0.114	0.027	0.063	0.170
VEI→SA→EA	0.120	0.028	0.070	0.179	0.114	0.030	0.055	0.175
VEI→PI→SA→EA	0.079	0.016	0.050	0.112	0.107	0.021	0.068	0.150
FIT→PI→EA	0.201	0.029	0.144	0.262	0.106	0.029	0.051	0.165
FIT→SA→EA	0.219	0.036	0.151	0.294	0.188	0.031	0.129	0.252
FIT→PI→SA→EA	0.059	0.015	0.033	0.091	0.056	0.014	0.032	0.086

## Data Availability

Exclude this statement.

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
