# Peer review of "Modeling Community Residents’ Exercise Adherence and Life Satisfaction: An Application of the Influence of the Reference Group"

_ijerph, 2022, doi:10.3390/ijerph192013174_

Round 1
Reviewer 1 Report
As a reviewer, I was presented with an article that had been very carefully considered from a structural point of view. Its structure is clear and the course of action presented in a way that the reader can understand. In my opinion, the authors have dealt sufficiently with the very large number of hypotheses that arose from the division adopted (life satisfaction, personal investment, strategic and cultural fit, etc.). The whole was 'encapsulated' by correctly selected research and statistical tools. However, I would like to draw attention to a few doubts I have after reading the article. In the first paragraph of the introduction of the claim about the importance of exercise during the COVID-19 pandemic, an article from 2016 is cited. This may raise unnecessary questions of logical construction.
The second issue is the unfortunate, in my opinion, definition of Fuzhou as a 'normal city' in section 2.1. In this sense, is Xiamen an abnormal city?
At the same time, I have a doubt as to whether the second, third and fourth paragraphs in section 2.5 should be there? Similarly, the last paragraph of section 6.
The literature has been aptly and reasonably selected. The only doubt might be about their 'age'. Only 34% of the items are from the last 5 years (2018 and newer).
Author Response
We are grateful for the reviewer’s valuable comments. According to the suggestions and comments of the reviewers, our paper has been greatly improved. Please see the attachment for details. If you have any questions, please do not hesitate to let us know.

Reviewer 2 Report
GENERAL COMMENTS
This is a large sample survey of factors relating to exercise behaviour in China. The authors attempt to highlight intervening factors that may relate to exercise adherence between two different centres (one urban, one more rural). Thorough analysis was done on more 48 items under 7 major constructs, although it is not clear to the reader how the individual items and constructs inform the conclusions of the paper. There is a need for editing of content for clarity of meaning, and a re-working of the discussion to relate back to the primary findings from the analyses.
For example, interpretation of meaning from the written English is difficult. Requires thorough review for English language writing. Eg. “formation mechanism”, “mechanism of the reference group and strategic? and cultural fit” – not sure what this is supposed to mean.
“ This study take(s) personal investment and life satisfaction as mediate variable(s) to explore the ‘formation mechanism’ of residents’ continuous participation in exercise based on SOR theory. – is this better worded as “This study evaluates the contribution of personal investment and life satisfaction as factors influencing exercise participation of residents, based on SOR theory.”
Eg. ‘Verify the chain mediating effects” ?
Many reference citations do not relate to the inference from the sentence. For example, 2nd para of intro, the statement is general to factors in exercise participation for all, but reference is specific to people with disabilities. These factors can be very different for these different groups. This is one example of the message-reference misalignment. Many sentences do not align with an appropriate citation. See example re: covid and PA. Not sure all of the ‘behavioural intention’ references to products relates to same constructs of behavioural intention for participation in exercise. Suggest a thorough examination of citations supporting statements in the paper.
The terms reference group, strategic and cultural fit seem abstract relative to their intent in the paper. After reading the paper, it is still not clear what these are meant to mean. Personal investment and life satisfaction are more concrete and relational. Strategic and cultural fit related in the cited research to the development of a marketing ‘brand’ – it is not given that this term translates the same way to exercise behaviour. It is somewhat justified in the 5.1 implications, but consider using different descriptors.
SPECIFIC COMMENTS
Intro. First statement is complex with focus on Covid-19 effects but references do not relate to Exercise during Covid-19 and its influence. Next line specifically on COVID and PA.
‘The reference group’ needs to be defined early and clearly. It is described later in the literature review, but for the reader it needs to be early, along with who the reference group is in this study.
Relative to hypotheses – PI, PF and PF are abstract concepts – your hypotheses should relate to specific variables that you measure (such as life satisfaction, personal investment, and similarity with reference group) as specific variables that relate to continuous exercise participation.
Abstract.
Spell out SOR theory in abstract first time. The abstract does not clearly describe the intent and outcomes of the study.
Lit background. 1.1.1 para 2. Reference group (PI) , PF and PV. Are these normally accepted acronyms for informational influence, utilitarian influence and value expressive influence respectively? Given that they don’t directly relate to the concepts, it is difficult to follow these relationships in the discussion with multiple acronyms relating to both individual variables and the higher order concepts.
1.1.2 high awareness precedes doing exercise regularly
1.1.4 ‘personal investment’ is also referred to as ‘outcome efficacy’ i.e. confidence that the outcome will be of benefit.
1.1.6 Stimulus-Organism-Response (S–O-R) theory refers that stimuli elements [clarify] affect individuals psychology, evaluation and cognitive [what?] towards things, and then influence individuals’ response (Jacoby, 2002). Cognitive toward things – their thoughts, understanding? intentions? Motivations?
Methods
2.1 Description of the sample as opposed to ‘sampling’ which is ‘Data collection’ covered in 2.4.
2.2. should have examples of some questions. And/or availability of the questionnaire on-line.
2.3 Reliability values very good.
2.4. How was the survey sent to participants? Was it voluntary? Or part of an incented program?
2.5 “It shows the participants … what is ‘It”. Table ? shouldn’t there be a table of demographic data?
This section is largely incomplete and contains a large body of text from the journal guidelines.
There is no statement on ethical approval.
3. Results
Values are all very high for a survey, probably due to the large sample size.
Extensive analysis performed.
4. Discussion. It is difficult to understand which relationships are ‘most’ important when everything seems to be significant in Table 4 It appears from Table 5 that only the PI-> MIà CP relationship is significant, but the discussion doesn’t seem to relate back to the most important results directly.
5.2 Not sure if the analyses performed substantiates the conclusions around cost to exercise. Was this specifically analyzed with this data? It is not clear how the conclusions made in 5.2 are directly supported by the data. Are each of these conclusions related to the analyses of personal investment and non-signfiicant differences in (MI) i.e education and income between the two cities? This connection to the findings needs to be stronger.
6. This section also has journal guideline info in the text.
References
Some journal titles are in abbreviated form, most are spelled out. Ensure consistency with Journal format guidelines. Eg. Ref # 5 no journal index or page numbers
Recommend review of all citations for formatting and relatability to statements in the text.
Author Response

(The authors gave the same response as above.)

Round 2
Reviewer 2 Report
The authors have acknowledged the reviewers concerns and made changes to the paper that improve its readability and presentation. The paper provides a reasonable analysis of a modest data set comparing factors related to ‘continuous exercise participation’. Although the authors state that a English readability edit has been done, there may still be some parts of the meaning or intent, that are being lost in translation and therefore it is recommend that a further or additional overall editing of the paper be done for clarity.
For example, the authors refer to ‘continuous participation in exercise’; after reviewing this paper again, the meaning of this in English is likely ‘adherence to exercise’ Or ‘exercise adherence’. This is a different implied meaning that simply ‘continuous participation’ and implies the mental/social constructs that the authors investigate in the paper. It is recommended that if this is the intent of this, that this be changed throughout.
Similarly the term ‘stimulate’ has a different connotation than I think the authors intend. A better term is likely ‘variable’, ‘contributor’ or ‘factor’ ‘facilitator’. External stimulus is a very direct application – usually in terms like a discrete stimulus such as a dose of exercise, or temperature exposure, when it is psychological and/or perceived, it is much more complex in a model.
As outlined below, there are also still some grammatical and typographical errors that exist that should also be reviewed.
Page 2 lines 63-66 . Therefore, based on SOR theory, residents’ behavior is originated influenced by an individuals’ evaluation of external stimulate stimuli and external stimuli are residents’ behavior influencing factors. However, little comprehensive research [exists] on how external variables influence people’s willingness to exercise on a regular basis
Pg3 lines 129-130… Therefore, intention to adhere to exercise is an indicator of exercise adherence.
Pg 5 lines 235-237 Stimulis elements are external environment factors; organism is individuals’ internal mental and psychological state like psychology, evaluation and cognitive towards things; response is individuals’ behavior outcomes which applied to stimulus and organism. ??? This sentence doesn’t make sense.
Pg 5 lines 312-315 We sent offline questionnaire in exercise places for voluntary, and sent online questionnaire through Wenjuanxing to the people who living in Fuzhou and Xiamen by people who have been surveyed. What is Wenjuanxing??
As ‘stimulations’
Eg. Pg 13 lines line 477 strategic and cultural fit as ‘stimulations’ will affect residents’ investments in exercise. ‘contributors’? ‘Factors’, facilitators?
Lines 482-483 the psychological fulfilment leads [to] residents’ life satisfaction. Finally, Therefore, residents are more likely to participate in exercise.
Author Response
Modeling Community Residents’ Exercise Adherence and Life Satisfaction: An Application of the Influence of the Reference Group
Response to Reviewer 2 (All the changes in “Track Changes” in the manuscript):
|
Remark |
Overall reaction |
Changes made |
|
For example, the authors refer to ‘continuous participation in exercise’; after reviewing this paper again, the meaning of this in English is likely ‘adherence to exercise’ Or ‘exercise adherence’. This is a different implied meaning that simply ‘continuous participation’ and implies the mental/social constructs that the authors investigate in the paper. It is recommended that if this is the intent of this, that this be changed throughout.
|
We are grateful for the reviewer’s valuable comments. We agree with the reviewer.
|
The full text uses adherence to exercise’ or ‘exercise adherence’ of ‘continuous participation in exercise’. |
|
Similarly the term ‘stimulate’ has a different connotation than I think the authors intend. A better term is likely ‘variable’, ‘contributor’ or ‘factor’ ‘facilitator’. External stimulus is a very direct application – usually in terms like a discrete stimulus such as a dose of exercise, or temperature exposure, when it is psychological and/or perceived, it is much more complex in a model. |
We agree with the reviewer. However, the research framework of the manuscript is based on Stimulus-Organism-Response (S–O-R) framework. SOR explains the formation framework of the individual's response to external stimuli. ‘stimuli’ is a proprietary term and an external variable. |
NA |
|
As outlined below, there are also still some grammatical and typographical errors that exist that should also be reviewed. |
We are grateful for the reviewer’s valuable comments. This manuscript was edited for proper English language, grammar, punctuation, spelling, and overall style by one or more of the highly qualified native English speaking editors at AJE. But, in order to improve the grammar accuracy, before publishing this manuscript, we will ask the editor to recommend grammar experts (https://www.mdpi.com/authors/english) to help revise the manuscript. |
NA |
|
Page 2 lines 63-66. Therefore, based on SOR theory, residents’ behavior is originated influenced by an individuals’ evaluation of external stimulate stimuli and external stimuli are residents’ behavior influencing factors. However, little comprehensive research [exists] on how external variables influence people’s willingness to exercise on a regular basis. |
We are grateful for the reviewer’s valuable comments. |
The reference group is the real or imagined person or group that has a significant influence on individuals’ values, attitudes, norms, and behaviors (Mi et al., 2019). As for exercise, it could be residents’ family, friends, colleagues, and neighbors.
Please see Page 2 lines 100-102.
|
|
Pg3 lines 129-130… Therefore, intention to adhere to exercise is an indicator of exercise adherence. |
We are grateful for the reviewer’s valuable comments. |
Based on the concept of behavioral intention and loyalty, exercise adherence (EA) is the residents’ behavior of doing exercise regularly and having a high awareness of ex-ercise. Therefore, the intention to adhere to participate is an indicator of continuous participation in exercise (Kim, 2022). Please see Page 3 lines 226-229.
|
|
Pg 5 lines 235-237 Stimulis elements are external environment factors; organism is individuals’ internal mental and psychological state like psychology, evaluation and cognitive towards things; response is individuals’ behavior outcomes which applied to stimulus and organism. ??? This sentence doesn’t make sense. |
We are grateful for the reviewer’s valuable comments. The statement is not clear, and we have deleted this sentence. |
|
|
Pg 5 lines 312-315 We sent offline questionnaire in exercise places for voluntary, and sent online questionnaire through Wenjuanxing to the people who living in Fuzhou and Xiamen by people who have been surveyed. What is Wenjuanxing?? |
We are grateful for the reviewer’s valuable comments. We revised it again. |
Investigators sent questionnaires to Fuzhou and Xiamen and delivered questionnaires both online and offline in consideration of COVID-19. We sent an offline questionnaire to exercise places for voluntary and collected online questionnaires to the people who live in Fuzhou and Xiamen by people who have been surveyed. Please see Page 5 lines 762-765.
|
|
Lines 482-483 the psychological fulfilment leads [to] residents’ life satisfaction. Finally, Therefore, residents are more likely to participate in exercise. |
We are grateful for the reviewer’s valuable comments. |
The reference group and strategy and cultural fit as external stimuli could affect res-idents’ attitude towards exercise, make residents know the importance and benefits of the exercise, to change personal investment in exercise. And then, as residents invest more in exercise, psychological fulfillment leads residents’ life satisfaction. Finally, residents are more likely to participate in the exercise. Therefore, personal investment and life satis-faction exert important roles in the process on the path of these antecedents on exercise adherence. please page 12. |